# Driving Forces in the Formation of Biocondensates of Highly Charged Proteins: A Thermodynamic Analysis of the Binary Complex Formation

**DOI:** 10.3390/biom14111421

**Published:** 2024-11-08

**Authors:** Matthias Ballauff

**Affiliations:** Institut für Chemie und Biochemie, Freie Universität Berlin, Forschungsbau SupraFab, Altensteinstrasse 23a, 14195 Berlin, Germany; matthias.ballauff@fu-berlin.de

**Keywords:** biocondensate, complex formation, counterion release, hydration

## Abstract

A thermodynamic analysis of the binary complex formation of the highly positively charged linker histone H1 and the highly negatively charged chaperone prothymosin α (ProTα) is detailed. ProTα and H1 have large opposite net charges (−44 and +53, respectively) and form complexes at physiological salt concentrations with high affinities. The data obtained for the binary complex formation are analyzed by a thermodynamic model that is based on counterion condensation modulated by hydration effects. The analysis demonstrates that the release of the counterions mainly bound to ProTα is the main driving force, and effects related to water release play no role within the limits of error. A strongly negative Δ*c_p_* (=−0.87 kJ/(K mol)) is found, which is due to the loss of conformational degrees of freedom.

## 1. Introduction

When positively charged polyelectrolytes interact with negatively charged polyelectrolytes in aqueous solution, this will result in a complex formation [1,2]. These “complex coacervates” have been the subject of intensive research, and surveys of the older literature may be found in various reviews [3,4]. Recently, this problem has found renewed interest because of the formation of biocondensates in living cells [5,6,7,8,9]. Here, coacervates are formed within cells through the interaction of anionic with cationic proteins [10,11]. The new interest in coacervates has led to a multitude of studies; details may be found in recent reviews [7,8,12]. Up to now, it was seemingly understood that charge–charge interaction may played a major role in the formation of biocondensates [7,13,14,15,16]. Simulations have led to a better understanding of the driving forces for biocondensate formation in charged systems [17,18,19]. Here, the question arises whether complex formation is driven by enthalpic or by entropic factors. Thus, the release of counterions [20,21] balancing the charge of macroions is expected to contribute to the gain of free energy during complex formation. In an important paper, Ou and Muthukumar [17] performed a comprehensive study of polyelectrolyte complexation by Langevin dynamics and found a significant enthalpic contribution at low to medium charge density. The entropic part of the free energy of binding only prevails at a sufficiently high charge density of the polyelectrolytes, as expressed through the linear charge density (Γ in ref. [17]). This finding has recently been criticized by Chen and Wang [22], who showed that a strong entropic contribution follows from the temperature dependence of the dielectric constant of water. They state that this electrostatic entropy, which is due to the reorganization of water dipoles, is the main driving force for complex formation, rather than the entropic contribution due to counterion release. Counterion release, on the other hand, has been identified as the main driving force for a number of systems for quite some time [20,21,23,24,25,26,27,28,29,30,31]. Moreover, the release of counterions has been observed directly by NMR techniques [32,33,34]. The work of Wang and coworkers [22], on the other hand, has underscored the importance of temperature as one of the decisive variables.

It is important to note that accurate data on the complex formation of flexible polyelectrolytes of opposite charge had already been obtained by Mascotti and Lohman many years ago [25,26,27,28]. In that work, the interaction of single-stranded RNA with oligolysines was analyzed by applying fluorescence techniques. The binding constant *K_b_* was measured at different concentrations of added salt *c_s_* at different temperatures. Thus, a full thermodynamic analysis could be carried out, leading to the result that the release of condensed counterions during complex formation is the main driving force for binding. A quantitative analysis of the data in terms of the counterion release model [20,35] was presented as follows: counterions are condensed to the highly charged polyelectrolyte and the fraction of counterions, thus bound to the macroion, and could be quantitatively modeled in terms of Manning’s theory [36]. Central to this approach is the definition of the charge parameter *ξ*:(1)$$\xi =\frac{\lambda_{B}}{b}$$
where *b* denotes the distance between the charge along the chain of the polyelectrolyte whereas *λ_B_* is the Bjerrum length (*λ_B_* = *e*^2^/4*πε*_0_*εkT*; *e*: elementary charge, *ε*_0_: permittivity of vacuum, *ε*: dielectric constant, *k*: Boltzmann constant, *T*: temperature). If *ξ* > 1, a fraction 1 − 1/*ξ* of counterions is condensed onto the polyelectrolyte chain. A part of these condensed counterions is released when the polyelectrolyte forms a complex with an oppositely charged protein. The gain of entropy provides a strong driving force for binding and the model predicts that log *K_b_* should scale with log *c_s_*. These predictions of theory were met with gratifying agreement when compared to the experimental data of various single-stranded RNAs with oligolysines varying in length [25,26,27,28]. Small deviations could be traced back and described quantitatively by effects due to hydration [25,26,27,28]. Hence, these thorough investigations strongly suggested that counterion release is the main driving force for the first step in biocondensate formation, namely the binary interaction of a cationic with an anionic polyelectrolyte. This finding is in full agreement with a more recent study by Priftis et al. using isothermal titration calorimetry [13].

The counterion release model can also explain the formation of complexes of long linear or branched polyelectrolytes with proteins [21]. Hence, natural polyelectrolytes, e.g., DNA or Heparin, can interact with patches of positive charge on the surface of the protein, which leads to a formation of a binary complex. Again, the log *K_b_* is found to scale linearly with log *c_s_* for many systems, as predicted by the counterion release model [21,29,37,38,39,40]. It should be noted that these considerations are directly supported by simulations on model systems [41,42] and by simulations using coarse-grained proteins interacting with polyelectrolytes [43]. Hydration effects that lead to slightly nonlinear plots of log *K_b_* vs. log *c_s_* can be incorporated using the fact that the complex formation is usually carried out at low concentrations of polyelectrolyte and protein [20,27]. Thus, the activity of water is bound to the activity of the salt ions by the Gibbs–Duhem relation [44]. Changes in hydration lead to a contribution that scales linearly with salt concentration and can be modeled [45] in terms of a parameter, Δ*w* [46,47,48,49]. A closed expression could be given, which comprises both the effect of counterion release as well as of hydration [50]. The model is based on the solute-partitioning model of Record and coworkers [51,52] which provides a quantitative treatment of Hofmeister effects. An application of this model [50] to the interaction of Heparin with lysozyme was recently presented and demonstrated that a comprehensive thermodynamic analysis of complex formation in solution can distinguish between the effects of counterion release and hydration [40].

An important experimental contribution to the analysis of biocondensates has been made by the Schuler group, who demonstrated that fluorescence techniques can be used for the study of complex formation down to the lowest concentrations [11,53,54]. Thus, Chowdhury et al. presented a comprehensive study of the formation of biocondensates using the IDPs prothymosinα (ProTα), carrying 44 negative charges, and linker histone H1, with 53 cationic charges [54]. The formation of complexes could be studied at the level of single molecules, which leads to the unambiguous determination of the binding constant of binary and ternary complexes. In this way, the binding constant of the binary complex formation can be obtained without the interference of a concomitant phase separation, which would set in when working at higher concentrations [13].

Here, the data of Chowdhury et al. [54] for the first step in biocondensate formation, namely the formation of a binary complex, will be evaluated. Binary complex formation is characterized by dissociation constants in the nanomolar region and thus provides a strong driving force for biocondensate formation. The present analysis will be carried out in terms of a purely thermodynamic model developed recently for the analysis of complex formation between proteins and polyelectrolytes [50]. Only the following two assumptions are necessary for this model: the effect of counterion release scales with the log of the salt concentration, whereas hydration effects scale linearly with salt concentration. The latter dependence is well established [20] and successfully used to analyze Hofmeister effects on proteins in solution [51]. It is important to note that the counterion release model employed here is based on experimental studies on the interaction of highly charged polyelectrolytes with their counter- and co-ions [55,56,57,58,59]. Hence, the decomposition of the thermodynamic data employed here is based on a firm experimental basis. The present analysis is therefore capable of identifying the main driving forces for the interaction of highly charged IDPs in solution and the parameters derived here provide a firm basis for a more detailed statistical–mechanical model.

## 2. Thermodynamic Analysis

All studies carried out so far on complexation rely on the mass action law, and the strength of binding can be expressed in terms of a measured binding constant *K_b_*, which is related to the free energy of binding Δ*G_b_*(*T*,c*_s_*) by
(2)$$∆G_{b}(T,c_{s})=-RTlnK_{b}$$
where the measured binding constant *K_b_* is defined through
(3)$$K_{b}=\frac{\left[PEP\right]}{\left[PEa\right][PEc]}$$
where [*PEP*], [*PEa*], and [*PEc*] denote the concentrations of the complex, the anionic, and the cationic polyelectrolyte, respectively. Following the procedure devised by Record et al. [20,35], the derivative of *K_b_* follows as [50]
(4)$$\frac{dlnK_{b}}{dln\;a_{\pm }}=∆n_{ci}-\frac{pm}{55.6}∆w+\frac{dln\frac{\gamma_{PEP}}{\gamma_{PEa}\gamma_{PEc}}}{d\ln a_{\pm }}$$

Here, *a_±_* is the mean activity of the salt ions and *p* = 2 for a monovalent salt with molality *m*. The first term on the right-hand side is the net number of anions and cations which either released or taken up during complexation. The second term is due to the Gibbs–Duhem relation and the hydration parameter Δ*w* measures the impact of the water molecules released or taken up when the complex is formed. The third term contains the activity coefficients of the complex and of both reaction partners. For the linear polyelectrolytes under consideration here, this term takes care of the Debye–Hückel contribution of the free counterions. Since this term is being considered properly (see the discussion of Equation (5) below), the activity *a_±_* can be replaced by the salt concentration *c_s_* in the system with full generality. Hence, all subsequent evaluations can be carried out using the concentrations of the components.

Evidently, Δ*n_ci_* constitutes the leading term in Equation (4), since the hydration term will only give an appreciable contribution if the molality *m* of the added salt is high. As a central step of the counterion release model, Δ*n_ci_* can now be written as [20,35]
(5)$$∆n_{ci}≅Z(1-\frac{1}{2\xi })$$

Here, *Z* denotes the number of charges involved in the binding of the complex. The expression in the bracket stems from part 1 − 1/*ξ* of the condensed counterions, plus term (2*ξ*)^−1^, which is due to the Debye–Hückel contribution of the ions (see Equation (4)) [20,35]. For sufficiently high charged polyelectrolytes, Δ*n_ci_* ~ *Z*. Then Δ*n_ci_* becomes a stochiometric coefficient and the release of the counterions can be described by a simple mass action law:(3a)$$K_{th}=\frac{\left[PEP\right]{{[M}^{+}]}^{{∆n}_{ci}}}{\left[PEa\right][PEc]}=K_{b}{\left[M^{+}\right]}^{∆n_{ci}}≅\;\frac{\left[PEP\right]{{[M}^{+}]}^{Z}}{\left[PEa\right][PEc]}$$ where *K_th_* is the thermodynamic binding constant and [*M*^+^] denotes the activity of the counterions [20]. Since the concentration of the polyelectrolyte is very small, [*M*^+^] can be equated to the salt concentration *c_s_* in the system. The insertion of Equation (3a) into Equation (2) then leads to the logarithmic dependence of Δ*G_b_*(*T*,*c_s_*) on *c_s_*. This argument shows that the counterion release model can be traced back in good approximation to the mass action law when the charge density *ξ* is high enough.

Equation (5) thus connects the Manning theory, and with this, general knowledge on the colligative properties of polyelectrolytes, to the problem of complex formation under consideration here. Hence, the general validity of Equation (5) can be discussed on the basis of the earlier experimental results on polyelectrolytes in solution as follows:(i)For small concentrations of the polyelectrolyte, the charge parameter *ξ* does not depend on the concentration of added salt *c_s_*. This is a basic assumption of the Manning theory and is well supported by experimental evidence already discussed by Manning [37]. The osmotic pressure measured in a system of a polyelectrolytes and added salt is given in very good approximation by the osmotic contributions of the free counterions of the polyelectrolyte and the salt ions [36] (additivity rule; see also the discussion by Alexandrowicz [55] and by Blaul et al. [58]). It is thus evident that the condensed counterions behave much in the way of a chemical bound species that does not contribute to the osmotic pressure in the system. The released counterions, on the other hand, do contribute to the osmotic pressure, and the mass action law applies for them (cf. Equation (3a)). Thus, the logarithmic dependence of the free energy of binding on salt concentration derives directly from this fact, we only need to take into account the effect of counterion condensation. As a consequence, Δ*n_ci_* does not depend on salt concentration and Equation (4) can be integrated.(ii)The charge parameter is *ξ*~(*εT*)^−1^ due to the definition of the Bjerrum length (see Equation (1)). Therefore, the decrease of the dielectric constant *ε* with temperature is mostly compensated, and the change in *ξ* in the usual range of experimental temperatures (5–50 °C) is very small. Thus, in excellent approximation, Δ*n_ci_* does not depend on temperature either. This agrees very well with a great number of experimental observations on, e.g., the interaction of DNA with various proteins [37] or on the interaction of highly charged systems with proteins in general [21,43,50]. Thus, a given system can be characterized by a single value of Δ*n_ci_* independent of temperature. By virtue of argument (i), Δ*n_ci_* is independent of salt concentration as well, and presents therefore the central parameter of this analysis.

Given these facts, it is evident that Equation (4) can be integrated and rendered in presence of monovalent salt ions [20,27,50].
(6)$$∆G_{b}\left(T,c_{s}\right)=RT\;∆n_{ci}lnc_{s}-RT\;0.036∆wc_{s}+∆G_{res}$$

The quantity $∆G_{res}$ describes the remaining part of $∆G_{b}\left(T,c_{s}\right)$ at a suitably chosen reference state [50]. Going along these lines, a closed expression that combines both the effects of counterion release and hydration can be developed [50]. Central to its derivation is the fact that $∆n_{ci}$ does not depend on temperature.
(7)$$∆G_{b}\left(T,c_{s}\right)=RT∆n_{ci}lnc_{s}+∆H_{0}-T∆S_{0}+({∆c_{p,0}+c}_{s}\frac{d∆c_{p}}{dc_{s}})\left[T-T_{0}-T\ln\left(\frac{T}{T_{0}}\right)\right]$$

The change in the specific heat Δ*c_p_* is another central parameter (cf. the discussion of the specific heat in ref. [60]) in this analysis and comprises the following two terms: First, the intrinsic Δ*c_p_*_,0_, which takes into account the changes in Δ*c_p_* due to the gain or loss of degrees of freedom during binding. Second, a term due to hydration that scales with salt concentration *c_s_*. Previous work devoted to complex formation of rodlike polyelectrolytes and rigid proteins showed that $∆c_{p,0}$ is negligible in these systems [50]. Here, we deal with highly flexible and disordered protein and Δ*c_p_*_,0_ is expected to be of appreciable magnitude (cf. also the discussion in ref. [61]).

A new characteristic temperature *T*_0_ has been introduced here, which describes the dependence of hydration on temperature through [50,61]
(8)$$∆w=\frac{\frac{d∆c_{p}}{dc_{s}}}{0.036R}\left(ln\frac{T}{T_{0}}+\frac{T_{0}}{T}-1\right)$$

For a positive coefficient $\frac{d∆c_{p}}{dc_{s}}$ the effect of hydration increases the magnitude of Δ*G_b_* for temperatures above and below *T*_0_. The parameters $∆H_{0}$ and $∆S_{0}$ denote the enthalpic and entropic contributions to the free binding at contact [50,62]. The residual free energy $∆G_{res}$ follows as [50,62]
(9)$$∆G_{res}=∆H_{0}-T_{0}∆S_{0}$$

It is interesting to note that Equation (7) resembles the well-known generalized van’t Hoff expression [63]
(10)$$∆G_{b}(T)=∆H_{b,ref}-T∆S_{b,ref}+∆c_{p}\left[\left(T-T_{ref})-Tln(\frac{T}{T_{ref}}\right)\right]$$
which can be re-written by use of *T_s_* as the reference temperature at which $∆S_{b}=0$ [50,64]
(11)$$∆G_{b}(T)=∆H_{b}(T_{s})+∆c_{p}\left[\left(T-T_{s})-Tln(\frac{T}{T_{s}}\right)\right]$$
where $∆H_{b}(T_{s})$ denotes the enthalpy of binding at *T_ref_* = *T_s_* [64]. The characteristic temperature *T*_0_ defined through Equation (7) equals *T_s_* if the term due to counterion release is vanishing. In this way, *T*_0_ becomes a parameter that measures the influence of hydration on complex formation.

The foregoing considerations suggest to analyze the experimental data in the following two steps: First, the dependence of $∆G_{b}$ on *c_s_* can be determined by Equation (6) neglecting the parameter Δ*w*. In this way, a good estimate of the parameters Δ*n_ci_* and $∆G_{res}$ can be obtained. Subsequently, the dependence of $∆G_{b}$ on *T* can be analyzed in terms of Equation (11) to obtain an estimate of $∆c_{p}$. Both steps proceed in a fully model-free fashion. In a second step, Equation (7) can be used to analyze $∆G_{b}(T,c_{s})$ for all data at once [40,50,64]. In this way, the thermodynamic information embodied in the present set of data can be assessed in a secure fashion.

## 3. Results and Discussion

The investigations of Chowdhury et al. [54] lead to the binding constants of the binary complex formation between the anionic IDP ProTα and the cationic IDP H1. Figure 1 displays the respective sequences of amino acids of both IDPs. In the case of ProTα, a highly inhomogeneous distribution is seen, that is, there is a long sequence consisting only of glutamic acid and aspartic acid, only interrupted by one or at most two uncharged amino acids. If we approximate the mean length of an amino acid by 0.35 nm, such an interruption of a charge sequence is smaller than the Debye length, which for high salt concentrations is still of the order of 0.8 nm and concomitantly larger for a smaller ionic strength. Hence, both sequences indicated in Figure 1a) can be treated as a single highly charged line that can be characterized by a local charge parameter *ξ* = 1.54. (cf. Equation (1)). For these sequences, the percentage of condensed counterions can be estimated to be 70% according to Equation (5). Thus, Δ*n_ci_* (Equation (5)) is expected to be ca. 19, i.e., 19 counterions are condensed and do not contribute to the osmotic pressure in the system. For the smaller sequence, *ξ* = 1.78, which is followed by Δ*n_ci_*~5. Thus, the arguments expounded in the section Thermodynamic Analysis can be fully applied here.

The increase in the strength of charge–charge interaction with increasing “blockiness” is well-borne out from recent simulations [7,30,65]. It should be noted, however, that the polyelectrolyte must exceed a certain length so that counterion condensation can take place. These effects related to the termini of the polyelectrolyte chains have been considered in detail by Manning [66] and seem to describe the experimental data obtained for short polyelectrolytes very well [67]. Thus, the effective length of a polyelectrolyte in which counterion condensation may take place follows by subtracting the Debye length of each end. Hence, Δ*n_ci_* (Equation (5)) is expected to be slightly smaller than the above estimate of 19. It should be kept in mind, however, that Δ*n_ci_* contains not only the release counterions of the anionic polyelectrolyte but also a number of the counterions of the cationic polyelectrolyte (see below).

It is interesting to compare this sequence of charged amino acids to the one found for the cationic IDP H1 (see Figure 1b). Here, it is obvious that there is no long virtually uninterrupted series of cationic amino acids; the charges are far more evenly distributed than is found for ProTα (Figure 1a). Typically, doublets or at most triplets of lysine are separated by one or two uncharged and hydrophobic amino acids. Such a sequence has been found earlier in proteins that interact strongly with Heparin (Cardin–Weintraub sequences [68,69]; cf. also ref. [70]). Thus, these sequences can interact closely with highly charged anionic polyelectrolytes. However, the correlation of the counterions to the macroion will be smaller for H1 than in case of ProTα. This can be estimated from the fact that the charge parameter *ξ* (cf. Equation (1)), as calculated for the entire chain, is only 0.58.

Evidently, the interaction of ProTα with H1 can be compared to the well-studied case of globular proteins binding to highly charged polyelectrolytes [71]. A highly charged negative polyelectrolyte such as, e.g., DNA or Heparin, with a large number of condensed counterions, interacts with small cationic patches localized on the surface of a protein (“Heparin binding site”; cf. ef. [70]). The number of positive charges of H1, however, is very high, and the question of why H1 exhibits no toxic interaction with cell organelles arises [72]. It has been known for a long time that the charge density of a cationic polyelectrolyte largely determines its toxicity [73]. This feature has been corroborated by more recent investigations and surveys of the problem [72,74]. One may speculate that the smaller correlation of the counterions may mitigate the intrinsic toxicity of H1 by preventing its unspecific interaction with cell membranes.

In the following, the binding constants obtained by Chowdhury et al. [54] for the binary complexes ProTα/H1 will analyzed in terms of Equations (6) and (10), as described above. Figure 2 displays the free energies of binary complex formation of ProTα/H1 as the function of salt concentration for three temperatures. The lines in this semilogarithmic diagram run parallel to each other. Similar findings have been made for many other systems [21,43,50], which directly demonstrate that Δ*n_ci_* does not depend on temperature. From the data measured at 295K, this parameter follows as Δ*n_ci_* = 18.2.

To the author’s best knowledge, this is the highest value ever measured for this parameter. It points to a very strong effect of counterion release, which is ultimately responsible for the huge binding constants measured at the lowest salt concentration for this system. Indeed, free energies of binding with a magnitude larger than 60 kJ/mol are enormous, given the fact that a complex formation of polyelectrolytes with proteins is usually characterized by values between 30 and 40 kJ/mol [21,71].

The results shown in Figure 2B of ref. [54] show two additional points of interest, as follows: 1. Experiments using different monovalent ions demonstrate the independence of the counterion release effect on the type of ions used in the experiments. This observation points to the fact that, here, we deal with purely electrostatic interactions that do not involve any specific interaction of the macroion with the monovalent counterions [50]. This finding points to the absence of hydration effects, and the analysis of the dependence of the free energy of binding on temperature will fully corroborate this result (see the discussion of Figure 3 below). 2. Figure 2B of ref. [54] displays data relating to divalent ions. In principle, such data should allow us to differentiate between the release of cations and anions: If only monovalent cations condensed to ProTα would be released, then the replacement of Na^+^-ions by Mg^2+^-ions would lead to a slope in plot of ln *K_D_* vs. ln *a_±_* in Figure 2B, which is half of the slope found for monovalent salt ions [20]. The slope seen in Figure 2B of ref. [54], both for experiments with Mg^2+^-ions as well as in presence of SO_4_^2−^ ions, suggests that approximately 13 ions are released. This finding could be explained by assuming that an equal number of positive and negative ions are released, which means that the number of released counterions will be reduced for salts like MgCl_2_ or K_2_SO_4_ by a factor of 0.75, as already noted by Chowdhury et al. [54]. The experimental accuracy of the data referring to divalent ions is smaller than the data referring to monovalent ions, however, and the comparison of theory and experiment is only semi-quantitative for this point. Experiments using mixtures of Mg^2−^ and Na^+^-ions would be very helpful in elucidating this point [75].

The dependence of the free energy of binary complex formation of ProTα/H1 temperature is shown in Figure 3. First, it is important to note that the curvature of these plots directly point to a negative specific heat of binding Δ*c_p_*. A quantitative analysis of this point by Equation (11) shows that Δ*c_p_*~−0.8 kJ/(K mol) for salt concentrations of 0.25 and 0.275 M, whereas a value of −2 kJ/(K mol) follows for 0.208 M. The problems of this analysis are obvious, however: the application of Equation (10) ultimately amounts to a numerical differentiation of Δ*G_b_*, and small experimental errors will lead to huge errors in Δ*c_p_*. Nevertheless, this analysis clearly reveals that Δ*c_p_* is negative.

In the second step, an in-depth analysis of the data displayed in Figure 2 and Figure 3 can now be carried out with the aid of Equation (7). Here, the entire set of data is analyzed at once using the MathLab routine *cftool*; that is, all 31 data points Δ*G_b_(T,c_s_)* at fitted by a single set of parameters [40,50,64]. At first, the hydration parameter *d*Δ*c_p_*/*dc_s_* is neglected and the data are fitted for a chosen value of the characteristic temperature *T*_0_. For *T*_0_ = 290 K, we obtain the following set of parameters: Δ*n_ci_* = 17.5; Δ*H*_0_ = 48.1 kJ/mol; Δ*S*_0_ = 0.107 kJ/(K mol), and Δ*c*_*p*,0_ = −0.87. The solid lines in Figure 2 and Figure 3 display the fit of all data by this set of parameters. The data set would be fully compatible with *d*Δ*c_p_*/*dc_s_* ≅ 1 kJ/(K mol M), which points to Δ*w*~+100, that is, to a release of water molecules at room temperature. However, the fit neglecting of this parameter, shown by the solid lines in Figure 2 and Figure 3, is fully sufficient. This is in full agreement with the fact that no ion-specific effects are seen (see above), which would point to Hofmeister effects due to hydration.

The fit of four parameters in total may appear questionable. However, Δ*n_ci_* is more or less identical with the value obtained directly from the plot in Figure 2, and Δ*c*_*p*,0_ agrees with the value estimated from the above analysis by Equation (10). So, we are left with the two remaining parameters Δ*H*_0_ and Δ*S*_0_ that lead to Δ*G_res_* = 16.8 kJ/mol according to Equation (9). The linear extrapolation of the plot shown in Figure 2 to *c_s_* = 1 M leads to Δ*G_res_* = 18.5 kJ/mol, which is nearby. This extrapolation works under the assumption that Δ*w* = 0, which seems to be fully supported by the above finding that *d*Δ*c_p_*/*dc_s_* can be neglected in good approximation.

The strong positive enthalpy Δ*H*_0_ = 48 kJ/mol compares favorably to the values of 37–58 kJ/mol measured directly by calorimetry [54]. In general, the interpretation of the measured enthalpies, however, should proceed with caution, as already remarked by Ou and Muthukumar [17]. The enthalpy measured in an ITC experiment may contain many contributions that are difficult to calculate and which may cancel out each other. In particular, it should be noted that the enthalpy Δ*H_ITC_* measured directly by ITC must not be confounded with the enthalpy of complex formation Δ*H_B_*. It is well-known that Δ*H_ITC_* may contain additional contributions from linked equilibria as in, e.g., protonation during complex formation. Thus, Baker and Murphy analyzed the binding enthalpies for biomolecular complex formation by comparison with the enthalpies of buffer ionization [76,77]. They could demonstrate that the linked equilibrium of buffer ionization can furnish a marked contribution to the observed enthalpy. Extrapolated to zero heat of buffer ionization, the measured binding enthalpies may even change their sign. The same observation was made Hileman et al. [78], or by La et al. more recently in a series of carefully conducted ITC experiments [79]. In principle, Δ*H_B_* can be derived from the measured Δ*H_ITC_* by extrapolation to a vanishing heat of buffer ionization. It also can be obtained through analysis of the experimental data with the aid of Equation (7), giving Δ*H*_0_ as the enthalpy at *T* = *T*_0_, or by Equation (11), which yields Δ*H_b_*(*T_s_*). Since the evaluation via Equation (7) used the entire set of data, the resulting Δ*H*_0_ is certainly the more accurate result. The resulting Δ*H*_0_ = 48 kJ/mol, hence, can be taken as a reliable result, which indicates that complex formation is accompanied by a strongly positive enthalpic contribution partially balanced by a marked positive entropic term *T*_0_Δ*S*_0_ = 31 kJ/mol. A similar observation has been made by Priftis et al. in their studies of complex coacervation of synthetic polyelectrolytes [13]. A possible explanation of these findings may be sought in the disturbance of the water structure by the released counterions which have formerly condensed onto the negative polyelectrolyte. A cleavage of hydrogen bonds will hence lead to a loss of enthalpy together with a concomitant raise of entropy because of the induced disorder. In general, a strongly positive enthalpic contribution Δ*H*_0_ together with a markedly positive entropy Δ*S*_0_ points to a disordering of water molecules upon formation of the complex. A similar situation has been discussed by Netz and coworkers when considering the hydration repulsion between biomembranes [80,81]. Here, too, it was found that at close distances a strong enthalpic repulsion occurs, which is not compensated by a concomitant entropic term. In addition to this, water polarization effects may enhance these effects [80,81]. The strong positive enthalpy may have its origin in terms related to the change in the structure of the water phase during complex formation.

Summarizing the above analysis, it is evident that the counterion release term with Δ*n_ci_* = 17.5 is dominating Δ*G_b_*. Thus, for the physiological salt concentration of 0.15 M, the first term on the right-hand side of Equation (7) amounts to −80 kJ/mol at 290 K, which demonstrates that counterion release is by far the strongest driving force for binary complex formation. At this point, it is interesting to discuss the entropic contributions in Equation (7) again and explicitly calculate the contribution to entropy from the dependence of the dielectric constant *ε* on temperature [22]. If the part of Δ*G_b_* due to the release of counterions is defined by
(12)$$∆G_{b,CR}\left(T,c_{s}\right)=RT\;∆n_{ci}lnc_{s}$$

The respective entropy deriving from this part is (cf. Equation (5))
(13)$$-∆S_{b,CR}=\left(\frac{\partial ∆G_{b,CR}}{\partial T}\right)=Rln\;c_{s}\left(∆n_{ci}+\frac{T}{2\xi }\frac{dln\;\lambda_{B}}{dlnT}\right)$$

In order to calculate the dependence of Bjerrum length *λ_B_* on temperature, we use the expression furnished for *ε(T)* by Malmberg and Maryott [82]. From these data, *dlnλ_B_*/*dlnT* follows as 0.3 for a temperature of 300 K. With a charge parameter *ξ* = 1.5 of ProTα, the correction to $∆S_{b,CR}$ given by the second term in the bracket is 0.1, which is much smaller than $∆n_{ci}$ = 17.5. This result shows that there is no significant part of the entropy, which is due to the dependence of the dielectric constant on temperature. It also demonstrates that the neglect of the dependence of Δ*n_ci_* on temperature (cf. the discussion of Equations (5) and (6) above) is fully justified.

Finally, we discuss the change in the specific heat Δ*c_p_*. Strong effects from hydration can be ruled out, since no ion-specific effects are observed for the present system, which would lead to a term depending on salt concentration (cf. the discussion of this problem in ref. [40]). The strongly negative Δ*c*_*p*,0_ = −0.87 kJ/(K mol) must therefore be traced back to the loss of conformational degrees of freedom of both binding partners during complex formation.

## 4. Conclusions

A phenomenological thermodynamic analysis of the binary complex formation of the highly positively charged linker histone H1 and the highly negatively charged chaperone Prothymosin α (ProTα) has been presented. This analysis is fully based on a model-free phenomenological approach that is in full accord with the well-known colligative properties of polyelectrolytes; no additional assumptions need to be invoked. We find that the release of counterions as expressed through the parameterΔ*n_ci_* (Equations (6) and (7)) is the main driving force for complex formation. In addition to this, a strong additional entropic term Δ*S*_0_ (Equation (7)) supports binding, whereas the enthalpic term Δ*H*_0_ (Equation (7)) derived from the above analysis is positive. Moreover, hydration effects as expressed through the term Δ*w* in Equation (6) were found to be small. Finally, the negative Δ*c*_*p*,0_ = −0.87 kJ/(K mol) points to a loss of conformational degrees of freedom of the IDPs in the complex, as expected. The entire analysis shows that the complex formation of the IDPs in solution may well be compared to the much-studied problem of proteins interacting with highly charged polyelectrolytes [21,71].

## Figures and Tables

**Figure 1 biomolecules-14-01421-f001:**
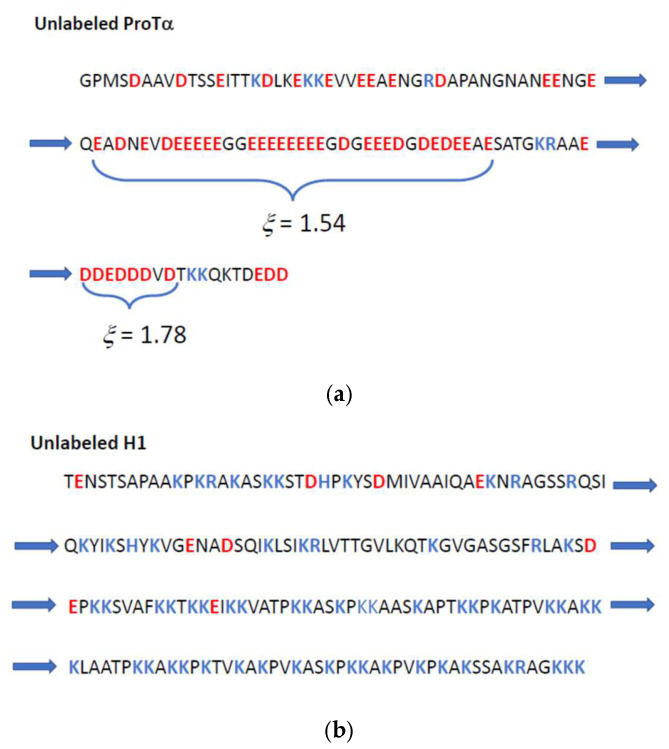
(**a**) The sequence of amino acids for the unlabeled anionic IDP ProTα, and (**b**) the sequence of amino acids of the unlabeled cationic IDP H1. Negatively charged amino acids are labeled in red, whereas positively charged amino acids are labeled blue. Highly charged sequences in ProTα (**a**) are characterized by the charge parameter *ξ* (cf. Equation (1)). The arrows indicate the continuation of the peptide chains.

**Figure 2 biomolecules-14-01421-f002:**
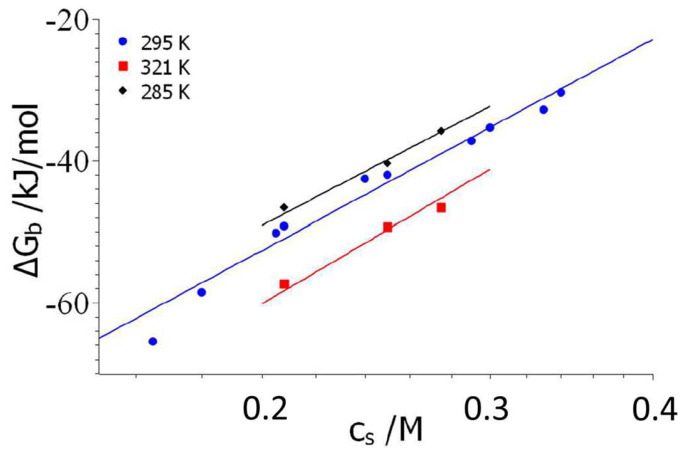
Semilogarithmic plot of the free energies of binary complex formation at a temperature of 285 K (black diamonds), 295 K (blue circles) and for 321 K (red quadrangles). Data taken from Figure 4a of Chowdhury et al. [54]. The solid lines show the fits according to Equation (7).

**Figure 3 biomolecules-14-01421-f003:**
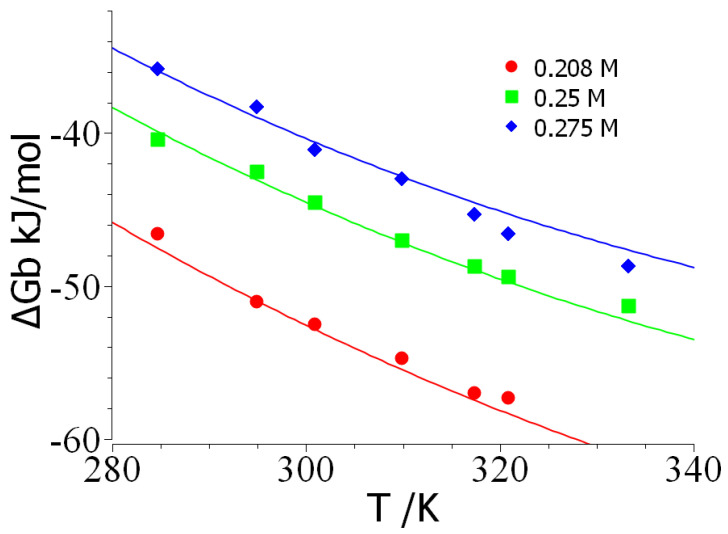
An analysis of the dependence of the free energy of binding Δ*G_b_* on temperature. The data have been taken from Figure 4E of Chowdhury et al. [54]. The solid lines mark the fit of the data by Equation (7).

## Data Availability

No new data were created or analyzed in this study. Data sharing is not applicable to this article.

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
