# Peer review of "Driving Forces in the Formation of Biocondensates of Highly Charged Proteins: A Thermodynamic Analysis of the Binary Complex Formation"

_biomolecules, 2024, doi:10.3390/biom14111421_

Round 1
Reviewer 1 Report
Comments and Suggestions for Authors
This is a thorough and scholarly quantitative analysis of the thermodynamics of a type of complex formation between two highly and oppositely charged protein molecules. The analysis was performed on published data in a paper by Chowdhury, et al (Proc Natl Acad Sci USA 2023, 120, e2304036120). As noted in the paper by Chowdhury, et al, "The association of polyelectrolytes with oppositely charged macromolecules is loosely referred to as polyelectrolyte complexation in the literature, which is commonly used as an overarching term that refers to the formation of small oligomers —especially dimers (soluble complexes)—but also dense liquid phases (coacervates) ..." That paper deals primarily with soluble dimer complexes, not liquid-liquid phase separation into a bulk coacervate phase. Therefore, that is the type of complexation that the Ballauff analysis addresses, and the Ballauff paper should make this clearer. Indeed, the term "biocondensates" in the title is a little misleading in this regard since the broad field of "biological condensates" refers more to intracellular liquid-liquid phase separation to form functional membraneless organelle compartments. Another aspect of where this paper would do well to clarify its limits is with respect to charge density. I believe that the Ballauff analysis pertains to a fairly high end of macromolecular charge density. If some quantitative limits could be put on the range of applicability of Ballauff's conclusion that counterion release dominates water re-organization it would be very instructive and helpful.
On a very mundane point, the references to Figures 1 and 2 in lines 329, 332, and 339, should be to Figures 2 and 3, respectively.
Author Response
First of all, I was pleased to see the positive reaction to my paper. To make it clearer that I only analyze the binary complex formation here, the title has been changed to
Driving Forces in the Formation of Biocondensates of Highly Charged Proteins: A Thermodynamic Analysis of Binary Complex Formation
Moreover, at the end of the introduction, I have re-worded the aim of the paper by
Here the data of Chowdhury et al. for the first step in biocondensate formation, namely formation of a binary complex , will be evaluated. Binary complex formation is characterized by dissociations constants in the nanomolar region and thus provides a strong driving force for biocondensate formation.
It should be clear by now that the paper deals with the first step of biocondensate formation.
As for possible limits of this analysis I was hesitant to add further comments on. The systems dealt with so far refer all to highly charged polyelectrolytes (see the discussion in ref. 53 of the paper). As the matter of fact, we are analyzing the complex formation of proteins with less charged polyelectrolytes as e.g. hyaluronic acid. At this point, however, it is too early to come up with quantitative conclusions.
The wrong number of the Figures in the text have been corrected – thanks for notifying me!
Reviewer 2 Report
Comments and Suggestions for Authors
The manuscript of M. Ballauff is an excellent thermodynamic analysis of existing experimental data obtained for histone H1 - prothymosine alpha complexation. The analysis combines classical counterion condensation principles with hydration effects. Results are in excellent agreement with existing experimental data.
The manuscript is well organized and written; it is also a mini review with rich literature overview.
I agree with the author regarding large binding free energies, which are several times larger in magnitude than those for protein/ion association. Dielectric constant of the medium not only changes with temperature but also with the composition of the solution, which can be even more important.
Some typos:
-equation 3a (line 146) appears after equation 5 (line 139)
-references to Figures 1 and 2 (line 332 and 336) should be to
Figures 2 and 3 instead?
Author Response
Thanks for the very positive comments and the benevolent reading of my paper!
The question of possible changes of the local dielectric constant with various parameter is important indeed. I have not found any clear experimental evidence yet which would tell us that is vastly different from its bulk value. Also, for the dilute systems under consideration here I would not expect any major change of this quantity. The present analysis would not be capable of analyzing any effect of this sort, of course. Since the present analysis works very well for a wide variety of systems (see the literature gathered here and the discussion in ref. 53), I am confident that these effects do not play a major role. More work is needed, preferably done on less charged polyelectrolytes.
The wrong number of the figures in the text have been corrected – thanks for indicating this problem! Eq.(3a) is just another version of eq.(3) and I have chosen the numbers to indicate this fact.
Reviewer 3 Report
Comments and Suggestions for Authors
The relatively recent discovery of “condensates” in the cytoplasm of biological cells has stimulated intense interest among biologists and physiologists, who however for the most part do not possess the tools to investigate these structures in the framework of state-of-the-art physical chemistry. The current paper under review here, written by Professor Ballauf who is expertly experienced in the complexation of proteins and polyelectrolytes, is welcome. Specifically, the paper presents a careful thermodynamic analysis of single-molecule energy transfer data from another team, Chowdhury et al, on the dimerization of the highly charged cationic disordered histone protein H1 with its highly charged anionic chaperone protein, also disordered, prothymosin-alpha. I agree with the implicit suggestion of Prof. Ballauf that Chowdhury’s analysis of the data can be improved and extended in important ways with the application of Ballauf’s own methodology, which is based on long established polyelectrolyte theory, in turn having long been documented by a large amount of experimental data. A very important aspect of Ballauf’s method is its ability to distinguish between the roles of counterion release and hydration changes on binding.
Ballauf’s data analysis establishes convincingly that the dimerization of these two disordered proteins is driven almost entirely by counterion release with hydration changes being almost insignificant. Additionally, heat capacity effects are assessed and interpreted.
A minor point, which nonetheless puzzles me, is the assertion that the amino acid sequence of H1 does not condense counterions, so that all of the condensed counterions, and hence all of the released ones, are on the chaperone protein. I don’t think it has been experimentally established that H1 is not associated with condensed counterions. There are experimental methods to detect condensed counterions, and it is too bad that they have not been applied to H1. A very simple test would be to measure the salt dependence of, for example, Mg++ binding to H1.
A related minor point is the apparent contradiction in the assertion on page 7, “We only need to assume that an equal number of positive and negative ions are released.” If no counterions are condensed on cationic H1, then how can any negative ions be released? And where does the number 0.75 for divalent ions come from? From simple considerations of charge neutrality, I would have thought that the number of divalent ions released would be half the number of univalent ones.
Author Response
It gives me great pleasure to acknowledge the benevolent reading of my paper! Referee #3 is obviously a renown expert in this matter and I was pleased to see that this referee has been convinced by my arguments. The points raised by referee #3 are very valuable and are related to each other.
- I admit that there are no condensed counterions on H1 was a bit too strong and referee #3 is quite right in stating that the extend of ion correlation must be clarified experimentally first. So I changed the wording on p. 7 to
However, the correlation of the counterions to the macroion will be smaller for H1 than in case of ProTa.a. This can be estimated from the fact that the charge parameter x (cf. eq.(1)) as calculated for the entire chain is only 0.58.
and on p. 8
One may speculate that the smaller correlation of the counterions may mitigate the intrinsic toxicity of H1 by preventing its unspecific interaction with cell membranes.
Evidently, this problem is worth pursuing.
- The salts used by Chowdhury et al. were 1:2 and 2:1-salts, namely K2SO4 and MgCl2. So, the argument given in the SI of the paper assumes that in presence of divalent salts we compensate the charge of only half of the released ions. Since we need only half of the divalent counterions, we get the factor 0.75 which seems to agree with the experiments (cf. Figure 2B of Chowdhury et al.). However, referee #3 is right in questioning this point. The experimental uncertainty of the data referring to divalent salts is much higher than for monovalent salts as is clear from inspection of Figure 2B of Chowdhury et al.. So the data of Chowdhury et al. are not sufficient to come to further conclusions.
In order to address this point properly, I have re-worked the entire paragraph on p. 8 below:
The results shown in Figure 2B of ref. [57] show two additional points of interest: 1. Experiments using different monovalent ions demonstrate the independence the counterion release effect on the type of ions used in the experiments. This observation points to the fact that here we deal with purely electrostatic interaction that do not involve any specific interaction of the macroion with the monovalent counterions. [53] This finding points to the absence of hydration effects and the analysis of the dependence of the free energy of binding on temperature will fully corroborate this result (see the discussion of Figure 3 below). 2. Figure 2B of ref. [57] displays data relating to divalent ions. In principle, such data should allow us to differentiate between the release of cations and anions: If only monovalent cations condensed to ProTa would be released, replacement of Na+-ions by Mg2+-ions would lead to a slope in plot of ln KD vs. ln a± in Figure 2B that is half of the slope found for monovalent salt ions. [21] The slope seen in Figure 2B of ref. [57] both for experiments with Mg2+-ions as well as in presence of SO42—ions suggests that ca. 13 ions are released. This finding could be explained by assuming that an equal number of positive and negative ions are released, which means that the number of released counterions will be reduced for salts like MgCl2 or K2SO4 by a factor of 0.75 as already noted by Chowdhury et al. [57]. The experimental accuracy of the data referring to divalent ions is smaller than the data referring to monovalent ions, however, and the comparison of theory and experiment is only semi-quantitative in this point. Experiments using mixtures of Mg2- and Na+-ions would be very helpful to decide this point. [78].
I hope that these changes of the manuscript meet fully the expectations of referee #3.